# Acetylcholine modulates prefrontal outcome coding during threat learning under uncertainty

Gaqi Tu[1,2], Peiying Wen[1], Adel Halawa[3], Kaori Takehara-Nishiuchi[1,2,4]*

[1]Department of Psychology, University of Toronto, Toronto, Canada; [2]Collaborative Program in Neuroscience, University of Toronto, Toronto, Canada; [3]Human Biology Program, University of Toronto, Toronto, Canada; [4]Department of Cell and Systems Biology, University of Toronto, Toronto, Canada

## eLife Assessment

This is an **important** study using a combination of optogenetics and calcium imaging to provide insight into the function of the cholinergic input to the prelimbic cortex in probabilistic spatial learning as it relates to threat. These data are timely in contributing to an ongoing discussion in the field about the role of phasic cholinergic signaling to the cortex, about which relatively little is known. The strength of the evidence is **incomplete** and could be improved by changes in task design and analyses, cross-validation of the conditions in calcium imaging, as well as the incorporation of control experiments to more definitively show it is indeed acetylcholine working in this circuit.

*For correspondence:
kaori.nishiuchi@utoronto.ca

**Competing interest:** The authors declare that no competing interests exist.

**Abstract** Outcomes can vary even when choices are repeated. Such ambiguity necessitates adjusting how much to learn from each outcome by tracking its variability. The medial prefrontal cortex (mPFC) has been reported to signal the expected outcome and its discrepancy from the actual outcome (prediction error), two variables essential for controlling the learning rate. However, the source of signals that shape these coding properties remains unknown. Here, we investigated the contribution of cholinergic projections from the basal forebrain because they carry precisely timed signals about outcomes. One-photon calcium imaging revealed that as mice learned different probabilities of threat occurrence on two paths, some mPFC cells responded to threats on one of the paths, while other cells gained responses to threat omission. These threat- and omission-evoked responses were scaled to the unexpectedness of outcomes, some exhibiting a reversal in response direction when encountering surprising threats as opposed to surprising omissions. This selectivity for signed prediction errors was enhanced by optogenetic stimulation of local cholinergic terminals during threats. The enhanced threat-evoked cholinergic signals also made mice erroneously abandon the correct choice after a single threat that violated expectations, thereby decoupling their path choice from the history of threat occurrence on each path. Thus, acetylcholine modulates the encoding of surprising outcomes in the mPFC to control how much they dictate future decisions.

## Introduction

In daily life, we rely on environmental cues to choose the course of action that would maximize positive outcomes and minimize adverse outcomes. This process requires tracking the discrepancies between received and expected outcomes, known as prediction errors (*Rescorla and Wagner, 1972*; *Sutton, 1988*; *Mackintosh, 1975*; *Pearce and Hall, 1980*). Prediction errors control how much a given outcome changes the expectation of subsequent outcomes, such that the larger the prediction error,

the greater the change. This process, however, becomes complicated when outcomes follow cues or actions unreliably (*Yu and Dayan, 2005*; *Bach and Dolan, 2012*; *Soltani and Izquierdo, 2019*). Under such noisy conditions, even correct cues and actions occasionally result in large prediction errors. Such anomalies may indicate the need to accelerate learning to cope with a sudden, drastic environmental change. Alternatively, they may fall within a typically expected range of outcome variability and should be dismissed.

These two conflicting interpretations of surprising outcomes lead to 'unexpected' and 'expected' uncertainty, respectively (*Yu and Dayan, 2005*; *Bach and Dolan, 2012*; *Soltani and Izquierdo, 2019*). Humans and rodents can use both types of uncertainty to adjust the learning rate properly, and this essential ability relies on neuromodulatory systems, such as serotonin (*Clarke et al., 2004*; *Boulougouris et al., 2008*; *Brigman et al., 2010*; *Matias et al., 2017*; *Iigaya et al., 2018*; *Grossman et al., 2022*), noradrenaline (*Rowe et al., 1996*; *Tait et al., 2007*; *Alexander et al., 2007*; *Mueller et al., 2008*; *Uematsu et al., 2017*; *Lapiz and Morilak, 2006*; *Marshall et al., 2016*; *Fitzgerald et al., 2015*), dopamine (*Floresco et al., 2006*; *Stopper et al., 2014*; *Jenni et al., 2017*; *St. Onge et al., 2011*; *Ragozzino, 2002*), and acetylcholine (ACh) (*Marshall et al., 2016*; *Witte et al., 1997*; *Chiba et al., 1999*; *Phillips et al., 2000*; *Vossel et al., 2014*; *Hassani et al., 2023*). Parallel evidence also suggests that adaptive learning and decision-making under uncertainty rely on the integrity of the anterior cingulate cortex in humans and monkeys (*Soltani and Izquierdo, 2019*; *Rushworth and Behrens, 2008*; *Shackman et al., 2011*; *Behrens et al., 2007*; *Soltani and Koechlin, 2022*) and its homolog, the medial prefrontal cortex (mPFC), in rodents (*St. Onge and Floresco, 2010*; *Paine et al., 2015*; *Zeeb et al., 2015*; *Orsini et al., 2018*). Neurons in these regions track outcome expectations by drastically changing the activity patterns depending on how reliably outcomes follow cues (*Matsumoto et al., 2003*; *Amiez et al., 2006*; *Takehara-Nishiuchi and McNaughton, 2008*; *Yan et al., 2021*; *Jezzini et al., 2021*) or actions (*Jacobs and Moghaddam, 2020*; *Passecker et al., 2019*; *Park and Moghaddam, 2017*; *Choi et al., 2023*). They also signal prediction errors by scaling the magnitude of responses to outcomes according to how surprising it was (*Amiez et al., 2006*; *Bryden et al., 2011*; *Hayden et al., 2011*; *Matsumoto et al., 2007*; *Seo and Lee, 2007*; *Hyman et al., 2017*; *Kehrer et al., 2024*). However, how these two systems interact to enable the proper handling of ambiguous outcomes remains unknown.

Here, we investigated the potential role of ACh in modulating prefrontal neural ensemble dynamics during adaptive learning and decision-making in a stable but probabilistic environment. We focused on ACh because (1) theories implicate ACh in computations of expected uncertainty (*Yu and Dayan, 2005* ; *Yu and Dayan, 2002*), (2) the mPFC receives extensive axonal projections of cholinergic cells in the basal forebrain (BF) (*Bloem et al., 2014*; *Gielow and Zaborszky, 2017*; *Tu et al., 2022*), and (3) these cholinergic cells emit phasic responses precisely time-locked to reward and punishment (*Tu et al., 2022*; *Hangya et al., 2015*; *Harrison et al., 2016*; *Teles-Grilo Ruivo et al., 2017*; *Guo et al., 2019*; *Laszlovszky et al., 2020*; *Robert et al., 2021*; *Hegedüs et al., 2023*). Here, combining in vivo one-photon calcium imaging of mPFC cells with optogenetic manipulations of local cholinergic terminals in mice, we have shown that threat-evoked phasic cholinergic signals control the content of prediction error coding in the mPFC and limit the learning from surprising threats that are within an expected range of variability.

## Results

### Phasic cholinergic signals modulate the development of adaptive threat expectations

To investigate how threat-evoked phasic cholinergic signals modulate adaptive threat learning and its neural correlate in the mPFC, we designed a probabilistic spatial learning task in which mice chose one of two paths associated with different probabilities of threats to obtain rewards. Hungry mice went back and forth between two reward sites at the opposite corners of a square-shaped maze (*Figure 1A*). They always received sugar pellets at the reward sites [$P(Reward)$ = 100%]; however, they also occasionally received an air puff toward the face/body in the middle of two paths connecting the reward sites (threat sites). On one of the paths (high-threat path, ht-p), an air puff was delivered 75% of the time [$P(Threat|ht\text{-}p)$ = 75%], whereas on the other path (low-threat path, lt-p), it was delivered 25% of the time [$P(Threat|lt\text{-}p)$ = 25%]. The mice received air puffs at the same probability on a given

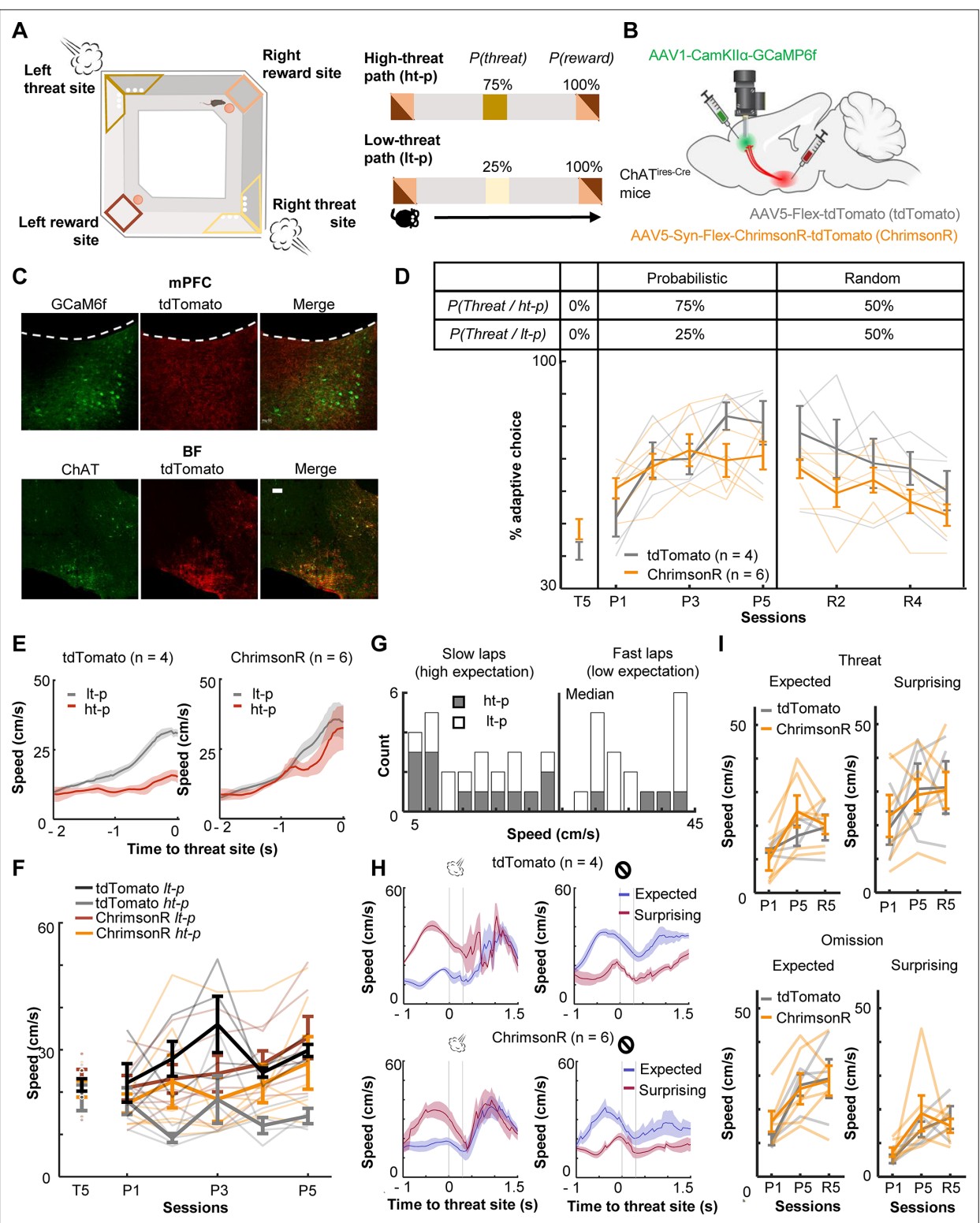

**Figure 1.** Optogenetic stimulation of basal forebrain–medial prefrontal cortex (BF–mPFC) cholinergic projections during threats impairs discrimination of two probabilistic outcome contingencies. (**A**) The maze used in the probabilistic spatial learning task (left). The probability of threat and reward assigned to each of two paths connecting two reward sites (right). Mice received air puffs 75% of the time on one of the paths (high-threat path) and 25% of the time on the other path (low-threat path). The color of the squares indicates the locations on the maze as specified in the illustration on the left. (**B**) The site of GRIN lens implantation and viral vector infusion. (**C**) Images showing ChrimsonR-tdTomato expressing cholinergic terminals (red) near the GCaMP6f-expressing neurons (green) in the mPFC (top). Images showing colocalization of ChAT and virally infected cells expressing tdTomato in

*Figure 1 continued on next page*

*Figure 1 continued*

the BF (bottom). The scale bar represents 100 μm. (**D**) The proportion of laps in which mice chose the low-threat path (% adaptive path choice, one thin line/mouse, mean ± SEM). The table on top indicates the assigned threat probability for each path. Adaptive path choice in the pre-training (T5) and random stage (R1–5) was defined as choosing the path that was the low-threat path during the probabilistic stage (P1–5). (**E**) The movement speed of mice while they ran toward the threat sites in P5 (mean ± SEM). (**F**) The approaching speed toward the threat sites on two paths (one thin line/mouse, mean ± SEM). The speed was averaged over a 500-ms window before the threat site entry. (**G**) The distribution of the speed approaching the threat site in all laps in P5 in a representative mouse. Laps with the approaching speed below or above the median were categorized as 'slow' and 'fast' laps, respectively. Laps on the high-threat path (gray) were likely to be slow laps, while laps on the low-threat path (white) were likely to be fast laps, consistent with differential threat expectations between the paths. (**H**) The movement speed around the threat sites in fast and slow laps in P5 (mean ± SEM). The speed in fast laps with threats and slow laps with omissions showed the mice's reaction to surprising outcomes (red). Conversely, the speed in slow laps with threats and fast laps with omissions showed their reaction to expected outcomes (blue). (**I**) The averaged speed during a 500-ms window starting from the threat site entry (one thin line/mouse, mean ± SEM). Laps were categorized into the combination of the approaching speed (fast or slow) and outcomes (threat or omission) and labeled as 'expected' and 'surprising' as defined in H.

The online version of this article includes the following source data and figure supplement(s) for figure 1:

**Source data 1.** The proportion of laps in which mice chose the low-threat path (% adaptive path choice).

**Source data 2.** The approaching speed toward the threat sites on two paths.

**Figure supplement 1.** Histological verifications and experimental design of the calcium imaging with optogenetic manipulation.

**Figure supplement 2.** Differentiation of movement speed depending on threat probabilities and expectations.

path regardless of which direction they ran. The timing of reward and threat delivery was controlled based on real-time tracking of mouse' position.

During the probabilistic spatial learning task, we conducted cellular-resolution calcium imaging from the prelimbic (PL) region of the mPFC with simultaneous optogenetic manipulations of the BF cholinergic terminals. We stimulated, rather than inhibited, cholinergic terminals for two reasons. First, because the relationships between ACh level and behavior follow an inverted U-shape, in which abnormally high ACh is as detrimental as abnormally low ACh (*Mineur and Picciotto, 2021*; *Cools and Arnsten, 2022*). Second, we have shown that the stimulation of threat-evoked phasic cholinergic signals in the PL prevents mice from associating the threat with a preceding cue, while their inhibition facilitates association formation (*Tu et al., 2022*). Thus, the stimulation, but not inhibition, of threat-evoked cholinergic signals disrupts prefrontal neural processing during threat learning.

To limit the expression of excitatory opsins to cholinergic cells, the viral-mediated gene transfer approach was applied to mice expressing Cre recombinase in cholinergic cells (ChAT[ires-Cre] mice; *Figure 1B*). The mice were injected with conditional adeno-associated viral vectors (AAVs) for expressing a red-shifted excitatory opsin, ChrimsonR, with a florescent reporter (ChrimsonR group, $n = 6$) or the reporter alone (tdTomato group, $n = 4$) in the horizontal diagonal band of Broca (HDB) region of the BF (*Figure 1B*, *Figure 1—figure supplements 1A, B*; based on our previous anatomical tracing result *Tu et al., 2022*). All mice were also injected with AAVs carrying a calcium indicator (GCaMP6f) under the control of a CaMKII promotor in the PL. Histological validation confirmed that the expression of ChrimsonR overlapped with cholinergic cells expressing ChAT, and their axons were located near putative excitatory cells expressing GCaMP6f in the PL (*Figure 1C*).

After the recovery period, all mice underwent 15 days of training, divided into three stages (*Figure 1—figure supplement 1D*), pre-training [T1–5; $P(Threat|ht\text{-}p) = 0\%$, $P(Threat|lt\text{-}p) = 0\%$], probabilistic [P1–5; $P(Threat|ht\text{-}p) = 75\%$, $P(Threat|lt\text{-}p) = 25\%$], and random stages [R1–5; $P(Threat|ht\text{-}p) = 50\%$, $P(Threat|lt\text{-}p) = 50\%$]. During the probabilistic and random stages, we stimulated local cholinergic terminals unilaterally by delivering orange LED stimulations (300 ms, 20 Hz, 8 mW/mm$^2$) to the imaged window. The LED stimulation was applied during ~50% of the air-puff delivery on both paths, allowing for monitoring threat-evoked activity with and without the LED stimulation (*Figure 1—figure supplement 1E*). During the pre-training stage (T5), both groups chose two paths with similar likelihood (*t*-test, $t(8) = 1.497$, $p = 0.173$; *Figure 1D*). When air-puff delivery and LED stimulation commenced (P1–5), tdTomato-expressing mice learned to choose the low-threat path preferentially (two-way mixed ANOVA, Session × Group interaction, $F(4,32) = 3.347$, $p = 0.021$; follow-up one-way repeated measure ANOVA, tdTomato, $F(4,12) = 12.469$, $p < 0.001$; *Figure 1D*). Initially, they approached both threat sites at comparable speeds (*Figure 1—figure supplement 2*). After learning, they ran more slowly on the high-threat path than the low-threat path (three-way mixed ANOVA, Group × Path interaction, $F(1,8) = 9.225$, $p = 0.016$; follow-up *t*-test, tdTomato, path

type, $t(38)$ = 5.576, p < 0.001; *Figure 1E, F*). In contrast, ChrimsonR-expressing mice continued to frequently choose the high-threat path even after 5 days of training (ChrimsonR, $F(4,12)$ = 1.837, p = 0.161; *Figure 1D*) and did not differentiate the running speed between the paths (ChrimsonR, path type, $t(58)$ = 1.495, p = 0.140; *Figure 1E, F*, *Figure 1—figure supplement 2A*). In the random stage (R1–5), both groups gradually reduced the probability of choosing the original low-threat path and no longer showed any differences in the path choice (Session × Group interaction, $F(4,32)$ = 0.588, p = 0.674; Session, $F(4,32)$ = 7.747, p < 0.001; Group, $F(1,8)$ = 2.269, p = 0.170.; *Figure 1D*) or speed (*Figure 1—figure supplement 2A*).

Because tdTomato-expressing mice learned to differentiate the speed toward the threat sites depending on threat probabilities (*Figure 1E*), we used the speed variations across laps to gauge the mice's expectation of threats: a slower speed indicated that they expected to receive threats, while a faster speed implied that they did not (*Figure 1G*). The comparison of the running speed between these two lap types revealed that the mice reacted differently to the same outcomes depending on their expectations. Specifically, mice accelerated more robustly after surprising threats than expected threats (*Figure 1H*, left). They also took longer to accelerate after surprising omissions than expected omissions (*Figure 1H*, right). Similar patterns were also detected in P1 and R5 (*Figure 1—figure supplement 2B*). In all these sessions, cholinergic terminal stimulation did not affect the reactions to expected or surprising outcomes (Threat: three-way mixed ANOVA, Session × Expectation × Group interactions, p = 0.285; Session, $F(2,16)$ = 5.626, p = 0.014; Expectation, $F(1,8)$ = 37.918, p < 0.001; Group, $F(1,8)$ = 0.039, p = 0.849; Omission: Session × Expectation × Group interactions, p = 0.360; Session, $F(2,16)$ = 12.840, p < 0.001; Expectation, $F(1,8)$ = 33.280, p < 0.001; Group, $F(1,8)$ = 0.156, p = 0.703; *Figure 1I*). These findings suggest that phasic cholinergic signals do not modulate the development of threat expectation but are essential for evaluating different probabilistic outcome contingencies.

## Enhanced phasic cholinergic signals augment the threat-evoked activity of PL cells

To probe neural mechanisms behind the observed behavioral effects, we used a head-mounted single-photon epifluorescence miniature microscope to monitor $Ca^{2+}$ activity of PL cells via a gradient-index (GRIN) lens (*Figure 2A*). The biological and optical crosstalk of optogenetic stimulation on calcium imaging is negligible (*Stamatakis et al., 2018*). We first investigated how cholinergic terminal stimulation affected the threat-evoked activity of PL cells by using the data during the first session with probabilistic air-puff delivery with LED stimulation (P1; *Figure 2B*). ~15% of PL cells increased $Ca^{2+}$ activity in response to the combined delivery of air puff and LED stimulation (*Figure 2C*; tdTomato, 14.7%; ChrimsonR, 17.9%). In the absence of LED stimulation, the magnitude of their puff-evoked activity was reduced in ChrimsonR-expressing mice but not in tdTomato-expressing mice (*Figure 2C*). Overall, the LED stimulation increased the puff-evoked activity more robustly in ChrimsonR-expressing than tdTomato-expressing mice (Kolmogorov–Smirnov test, w/ LED, p = 0.016; *Figure 2D*). This group difference was not detected without the LED stimulation (w/o LED, p = 0.386), confirming that the enhanced puff-evoked activity was due to the stimulation of the BF cholinergic terminals.

We also examined the effect of cholinergic terminal stimulation on PL cell activity in the absence of air puffs. In one pre-training session (T3), LED stimulation was applied when mice entered the corner of the maze, a location that would become the threat site in subsequent days (*Figure 2—figure supplement 1A*). Some cells changed their activity at the corner, and the LED stimulation disrupted the location-selective activity. However, the degree of the LED-induced change was comparable between the groups (Kolmogorov–Smirnov test, p = 0.237; *Figure 2—figure supplement 1B*). Thus, the optogenetic stimulation barely affected PL cell activity unless it coincided with the natural activation of cholinergic cells by threats (*Tu et al., 2022*; *Hangya et al., 2015*).

## Enhanced phasic cholinergic signals suppressed learning-dependent changes in outcome-related PL cell activity

Prefrontal cells are known to change responses to outcomes throughout learning, indicating that they encode values of expected outcomes (*Schoenbaum et al., 1998*; *Wallis and Miller, 2003*; *Padoa-Schioppa and Assad, 2006*; *Kim et al., 2008*; *Sul et al., 2010*) and the discrepancies between expected and actual outcomes (i.e., prediction errors) (*Amiez et al., 2006*; *Bryden et al., 2011*;

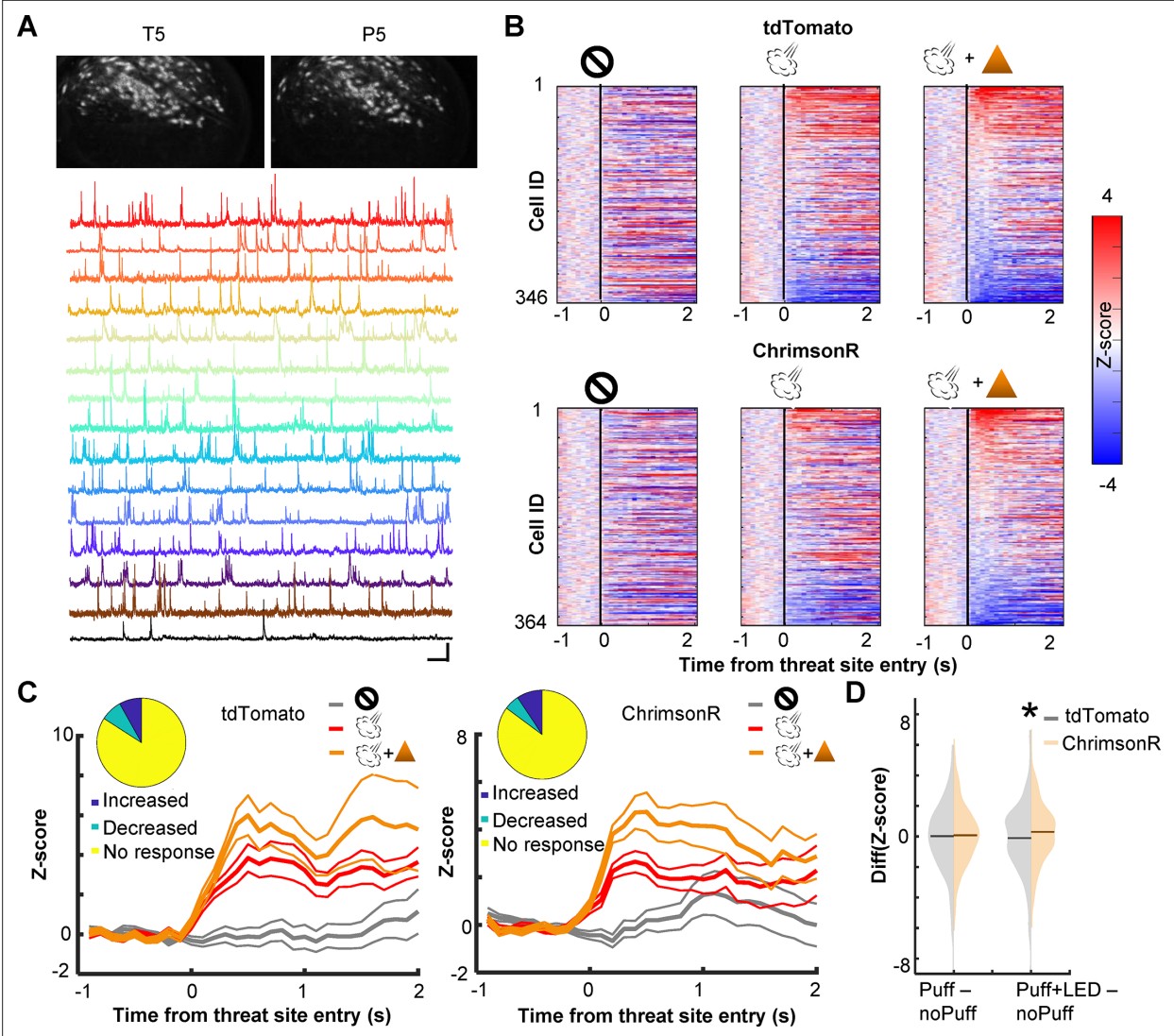

**Figure 2.** Cholinergic terminal stimulation augments PL cell activity in response to threats. (**A**) Field of views in a representative mouse from the last pre-training session (T5) and the fifth probabilistic session (P5; top). White contours represent registered cells of the corresponding session. Calcium traces of extracted cells (bottom). Scale bars represent 50 s × 5 z-score. (**B**) Pseudocolor plots showing the z-normalized cell activity aligned to the threat site entry during puff-omission (left), puff (middle), and puff and LED (right) laps. In all panels, cells were sorted by the response magnitude in puff and LED laps. (**C**) Z-normalized cell activity aligned to the threat site entry of threat-responding cells (mean ± SEM). (Inset) The proportion of cells with increased activity (blue) and those with decreased activity (green). (**D**) The distribution of the change in cell activity evoked by air puffs or air puffs and LED. Horizontal lines show the median. Z-normalized activity was averaged over a 500-ms window starting from the threat site entry in each lap type. The average activity in omission laps was subtracted from the average activity in puff laps or puff and LED laps. *p < 0.05.

The online version of this article includes the following figure supplement(s) for figure 2:

**Figure supplement 1.** Effects of cholinergic terminal stimulation on location-selective activity of PL cells.

*Hayden et al., 2011*; *Matsumoto et al., 2007*; *Seo and Lee, 2007*; *Hyman et al., 2017*; *Kehrer et al., 2024*). Therefore, we next asked how PL cells changed outcome-related activity with learning and how this process was affected by cholinergic terminal stimulation. We categorized laps into four types based on which path the mice took and whether they received an air puff. For laps with air-puff delivery, we only used ones without LED stimulation to isolate the learning-dependent changes from the direct effects of LED stimulation (*Figure 2C, D*). In each lap, the cell activity was aligned to the moment when the mouse's position tracking entered the threat site. In the first (P1) and last (P5) sessions of the probabilistic stage, a sizable proportion of PL cells changed their activity in response to threat delivery or omission (*Figure 3A, D*). In both groups, cells responding to outcomes on the

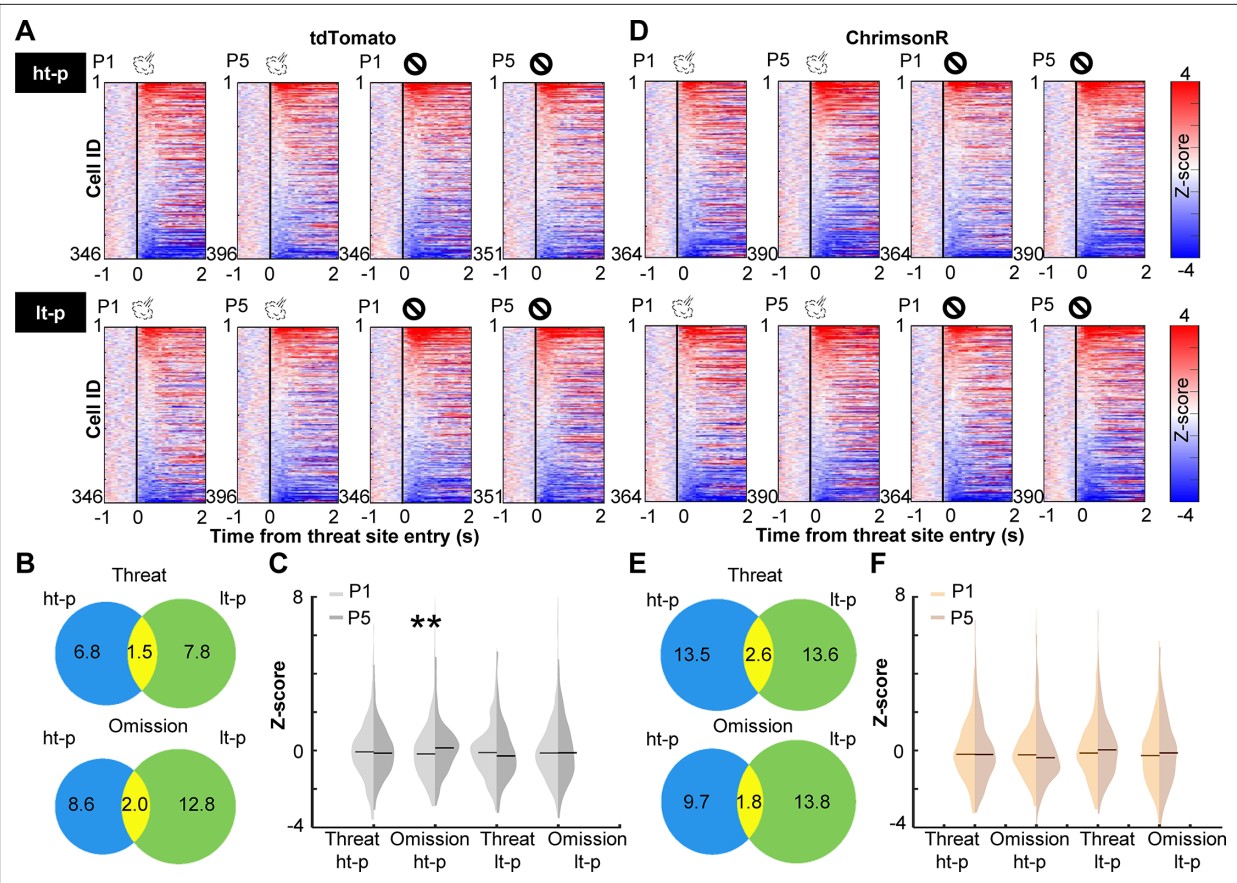

**Figure 3.** Cholinergic terminal stimulation abolishes learning-induced strengthening of PL cell responses to rare threat omission. (**A**) The changes in PL cell activity upon the threat site entry in tdTomato mice. Pseudocolor plots showing the z-normalized cell activity aligned to the threat site entry during the first (P1) and last session (P5) of the probabilistic stage. Laps were categorized into four types based on the path (high-threat path, ht-p; low-threat path, lt-p) and the threat state (threat delivery or omission). Cells were sorted by their activity in the threat site in each panel. (**B**) Venn diagrams depicting the proportion of cells responding to threats and their omissions on the ht-p (blue) and the lt-p (green) as well as their overlap (yellow) in P5. The numbers show the percentage of cells in each category. (**C**) The distribution of z-normalized activity in the threat sites in P1 (light gray) and P5 (dark gray) with horizontal bars depicting the median. (**D–F**) The same as A–C for PL cell activity in ChrimsonR-expressing mice. **p < 0.01.

The online version of this article includes the following figure supplement(s) for figure 3:

**Figure supplement 1.** Learning-dependent changes in outcome-related activity of PL cells.

high-threat path were distinct from those on the low-threat path (P1, *Figure 3—figure supplement 1A*; P5, *Figure 3B, E*), suggesting that cholinergic terminal stimulation did not affect the differentiation of outcome representations between two threat sites. In tdTomato-expressing mice, the omission-evoked activity on the high-threat path was shifted toward excitation from P1 to P5, while the threat-evoked activity was unchanged (Kolmogorov–Smirnov test, threat on ht-p, p = 0.485; omission on ht-p, p = 0.005; *Figure 3C*). Threat- or omission-evoked activity on the low-threat path was not changed across sessions (threat on lt-p, p = 0.106; omission on lt-p, p = 0.718). In ChrimsonR-expressing mice, the distribution of neither threat- nor omission-evoked responses was changed from P1 to P5 on either path (threat on ht-p, p = 0.265; omission on ht-p, p = 0.055; threat on lt-p, p = 0.279; omission on lt-p, p = 0.473; *Figure 3F*). The comparisons between the groups revealed that cholinergic terminal stimulation had no effects on threat- or omission-evoked activity in P1 (threat on ht-p, p = 0.731; omission on ht-p, p = 0.682; threat on lt-p, p = 0.404; omission on lt-p, p = 0.142; *Figure 3—figure supplement 1B*, left); however, it blocked the learning-dependent increase in the responses to rare threats omissions on the high-threat path (omission on ht-p, p < 0.001; *Figure 3—figure supplement 1B*, right). Thus, enhanced phasic cholinergic signals do not change how PL cells respond to subsequent outcomes within minutes; instead, they modulate the gradual restructuring of outcome representations over days.

## Phasic cholinergic signals augmented the selectivity of PL cells for signed prediction errors

The strengthening of cell activity to rare threat omission alluded to the sensitivity of PL cells to the discrepancy between expected and actual outcomes, that is, prediction errors. To further address this point, we investigated whether PL cells differentiated the outcome-related activity depending on the lap-by-lap fluctuations in threat expectations. Using the speed toward the threat site (*Figure 1G*), laps with air-puff delivery were split into surprising (faster speed) and expected (slower speed) threat laps. In parallel, laps with air-puff omission were split into surprising (slower speed) and expected (faster speed) omission laps. We then quantified the degree to which individual cells differentiated the outcome-related activity depending on expectations by subtracting the activity for expected outcomes from the activity for surprising outcomes. We observed strong differentiations of threat- and omission-evoked activity in all three sessions (*Figure 4A*). In both groups, cells showed diverse patterns of differential activity (*Figure 4B*). Some cells increased the activity in response to surprising threats but decreased the activity in response to surprising threat omission (*Figure 4C*, Up–Down), while others showed reversed patterns (Down–Up). Thus, these cells changed the response direction based on whether outcomes were better or worse than expected, a signature of signed prediction errors. In parallel, other cells showed the same direction of responses to surprising threats and surprising omission (Up–Up, Down–Down), suggesting that they were sensitive to unsigned prediction errors. In parallel, other cells showed expectation-dependent differentiations only to threats or their omissions (*Figure 4—figure supplement 1*). In P1, the proportion of cells encoding signed prediction errors was comparable to that encoding unsigned prediction errors (*Figure 4D*, left). These cells were much fewer than others that encode errors of one outcome type. All these cell types were detected in both groups in similar proportions. In P5, however, cholinergic terminal stimulation increased the proportion of cells encoding signed prediction errors (binominal test, p < 0.05; *Figure 4D*, middle). As a result, cells that encoded signed prediction errors became twice as common as those that encoded unsigned prediction errors. The overall proportion of cells encoding prediction errors was slightly increased. Similar patterns were also detected during R5 (binominal test, p < 0.05; *Figure 4D*, right). These findings suggest that enhanced phasic cholinergic signals augmented the encoding of signed prediction errors in the PL.

## Enhanced phasic cholinergic signal impairs adaptive learning of uncertain, but not certain, threats

The analyses of cell activity revealed that enhanced threat-evoked cholinergic signals modified the encoding of prediction errors in the PL. To further investigate how such changes in PL outcome coding affected behavior, we conducted follow-up behavioral experiments with bilateral excitation of the BF cholinergic terminals in the PL. AAVs were infused into the HDB of ChAT-cre mice to express either channelrhodopsin (ChR2, *n* = 10), ChrimsonR (*n* = 8) with control fluorophore, or control fluorophore alone (No-opsin, *n* = 14) in cholinergic cells (*Figure 5A, B*, *Figure 5—figure supplement 1A*). Their terminals in the PL *Figure 5—figure supplement 1B* were excited during every air-puff delivery in all sessions of the probabilistic paradigm [*Figure 5C*; *P(Threat|ht-p)* = 75%, *P(Threat|lt-p)* = 25%]. Consistent with the effect of the unilateral manipulation (*Figure 1D*), enhanced threat-evoked cholinergic terminal activity impaired the development of preference for the low-threat path (two-way mixed ANOVA, Session × Group interaction, $F(8,116)$ = 2.249, p = 0.029; follow-up one-way repeated measure ANOVA, No opsin, $F(4,52)$ = 11.190, p < 0.001; ChR2, $F(4,36)$ = 0.627, p = 0.646; ChrimsonR, $F(4,28)$ = 0.856, p = 0.502; *Figure 5D, E*). Because the task performance of the ChR2 and ChrimsonR groups was comparable, these groups were collapsed into a single group (opsin group) in subsequent analyses. Closer examinations of the movement pattern revealed that mice in the no-opsin group took longer to reach the threat site on the high- than the low-threat path (three-way mixed ANOVA, Path × Group × Session interaction, $F(4,96)$ = 0.508, p = 0.730; Path × Group interaction, $F(1,24)$ = 13.54, p = 0.001; follow-up t-test, no-opsin, path type, $t(108)$ = 3.979, p < 0.001; *Figure 5—figure supplement 2A*). In contrast, mice in the opsin group took comparable time on both paths (path type, $t(148)$ = 0.570, p = 0.569; *Figure 5—figure supplement 2A*). In contrast, the time taken from the threat site to the reward site was similar between the groups, and it became shorter across the sessions (Session, $F(4,96)$ = 5.091, p < 0.001; Group, $F(1,24)$ = 3.846, p = 0.062; Path, $F(1,24)$ = 0.297, p = 0.591; *Figure 5—figure supplement 2B*). Similarly, both groups initially increased and later decreased the

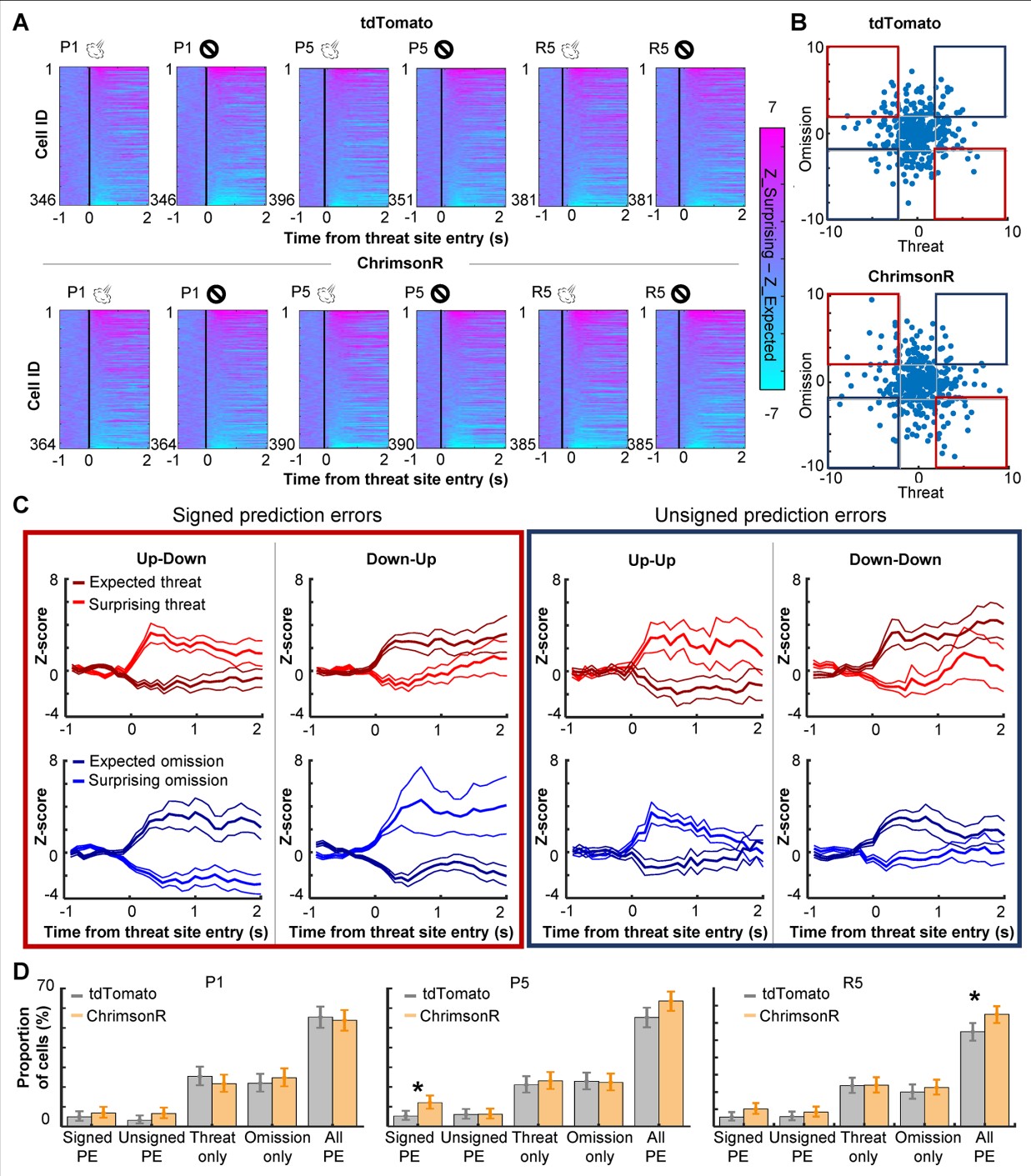

**Figure 4.** PL cells differentiate outcome-related activity depending on outcome expectations. (**A**) Pseudocolor plots showing the differentiation of outcome-related activity between expected and surprising outcomes during the first (P1) and last session (P5) of the probabilistic stage. The color depicts the difference in z-normalized cell activity between laps with expected outcomes and lap with surprising outcomes. Cells were sorted by the activity difference in the threat site in each panel. (**B**) Expectation-dependent differentiation of cell activity in response to threats (x axis) and their omission (y axis; one dot per cell) in P5. The z-normalized activity was averaged during a 500-ms window starting from the threat site entry. The averaged values in laps with expected outcomes were subtracted from those with surprising outcomes (differential activity). Blue squares highlight cells with the same response direction to surprising threats and omissions (unsigned prediction errors). Red squares highlight cells with an opposite response direction to surprising threats and omissions (signed prediction errors). (**C**) The averaged z-normalized activity across cells with different types of activity differentiation (mean ± SEM). A cell was selected when its differential activity for threats or their omission was greater than 2 (Up) or smaller than –2 (Down). Cells with significant differential activity for both threat delivery and omission were then further categorized into four types depending on their

*Figure 4 continued on next page*

*Figure 4 continued*

response direction. The activity of cells with significant differentiation only for one outcome type was depicted in *Figure 4—figure supplement 1*. (**D**) The proportion of cells in various differential activity types. Cells selective for outcome expectations were categorized into four types as defined in C and *Figure 4—figure supplement 1*. Error bars show the upper and lower confidence limit ($\alpha$ = 0.05). PE, prediction errors. R5, the last session of the random stage. *$p < 0.05$.

The online version of this article includes the following figure supplement(s) for figure 4:

**Figure supplement 1.** Expectation-dependent differentiation of outcome-related activity of PL cells.

**Figure supplement 2.** Lack of omission-evoked activation of cholinergic terminals in the PL.

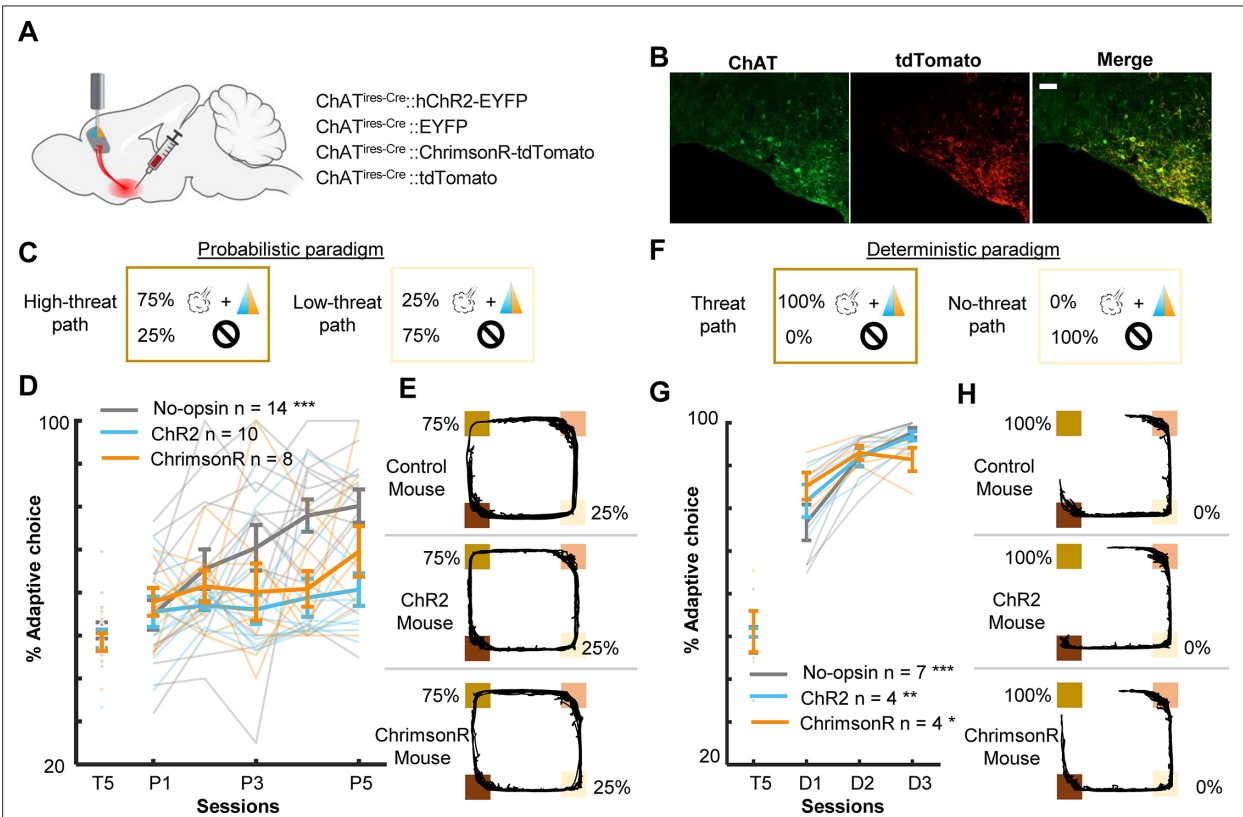

**Figure 5.** Threat-evoked phasic cholinergic signals constrain learning from surprising outcomes in a stable but probabilistic environment. (**A**) Sites of viral vector infusion and optic fiber implantation. (**B**) Representative images showing co-localization of ChAT and virally infected cells expressing tdTomato. The scale bar represents 100 µm. (**C**) The effect of cholinergic terminal stimulation on mice's behavior during the probabilistic paradigm. Schematic representation of threat probability and light stimulation on each path. LED stimulation was applied to the PL to activate the basal forebrain (BF) cholinergic terminals during every threat delivery. (**D**) The percentage of laps in which mice chose the low-threat path. (**E**) Movement trajectories of a representative mouse from each group. The locations of threat and reward sites were specified with the same color scheme as in *Figure 1A*. (**F–H**) Same as C–E for the deterministic paradigm. *$p < 0.05$, **$p < 0.01$, ***$p < 0.001$.

The online version of this article includes the following source data and figure supplement(s) for figure 5:

**Source data 1.** The percentage of laps in which mice chose the low-threat path in the probabilistic paradigm.

**Source data 2.** The percentage of laps in which mice chose the low-threat path in the deterministic paradigm.

**Figure supplement 1.** Locations of viral spread and optic fiber implants in mice underwent bilateral optogenetic manipulations.

**Figure supplement 2.** Movement patterns and the design of correlation analysis.

**Figure supplement 2—source data 1.** Movement patterns on the high- and low-threat path during the last pre-training session and all probabilistic sessions.

**Figure supplement 2—source data 2.** Movement patterns on the path with threats and path without threats during the last pre-training session and all deterministic sessions.

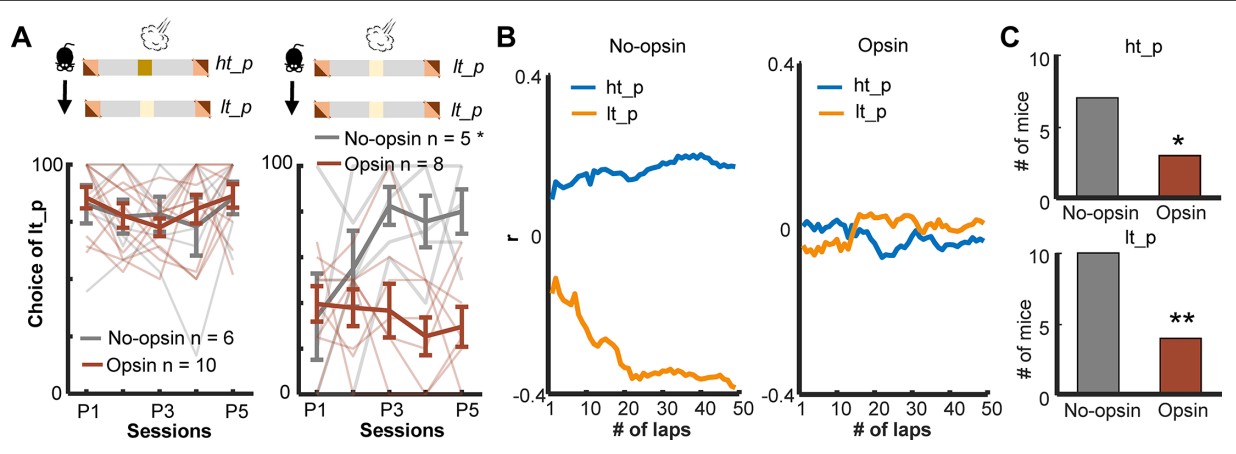

**Figure 6.** Enhanced threat-evoked phasic cholinergic signals impair the integration of outcome information over time. (**A**) The probability of choosing the low-threat path (lt-p) after an air-puff delivery on the high-threat path (ht-p; left; one thin line/mouse, mean ± SEM). Only mice with usable data in all five sessions were used for this analysis. (**B**) Correlation coefficients between the actual path choice and threat probability calculated with the different number of previous laps from representative mice. The greater positive values for the ht-p and negative values for the lt-p indicate stronger correlations with the estimated threat probability (see **Figure 5—figure supplement 2G**). (**C**) Numbers of mice showing significant correlations between the estimated threat probability and the choice of the ht-p (top). *p < 0.05, **p < 0.01.

The online version of this article includes the following source data for figure 6:

**Source data 1.** The probability of choosing the low-threat path after an air-puff delivery on the high-threat path.

time to initiate a subsequent lap (Session, $F(44,96) = 7.474$, p < 0.001; Group, $F(1,24) = 0.529$, p = 0.474; Path, $F(1,24) = 0.653$, p = 0.427; **Figure 5—figure supplement 2C**).

As outlined in the introduction, the control of learning rate is particularly essential when threats occur unreliably. To test whether cholinergic modulation of the PL is vital for probabilistic threat learning or threat learning in general, we conducted the same experiment with the deterministic paradigm [$P(Threat|ht-p) = 100\%$, $P(Threat|lt-p) = 0\%$]. We observed that all three groups preferred the path without threats (no-threat path) over the path with threats (threat path; two-way mixed ANOVA, Session × Group interaction, $F(4,24) = 3.195$, p = 0.031; follow-up one-way repeated measure ANOVA, no-opsin, $F(2,12) = 25.83$, p < 0.001; ChR2, $F(2,6) = 14.39$, p = 0.005; ChrimsonR, $F(2,6) = 7.199$, p = 0.025; **Figure 5D**). In addition, both groups took longer to reach the threat site on the threat than no-threat path (three-way mixed ANOVA, Session, $F(1,13) = 0.034$, p = 0.857; Group, $F(1,13) = 0.758$, p = 0.399; Path, $F(1,13) = 26.38$, p < 0.001; **Figure 5—figure supplement 2D**). They also took longer to reach the reward site on the no-threat than threat path (Session, $F(1,13) = 0.008$, p = 0.930; Group, $F(1,13) = 0.025$, p = 0.877; Path, $F(1,13) = 73.320$, p < 0.001; **Figure 5—figure supplement 2E**). On the contrary, both groups increased the time to initiate a subsequent lap after choosing the no-threat path (Session, $F(1,13) = 5.182$, p = 0.040; Group, $F(1,13) = 0.089$, p = 0.770; Path, $F(1,13) = 48.28$, p < 0.001; **Figure 5—figure supplement 2F**).

Intact learning in the deterministic paradigm (**Figure 5D**) suggests that enhanced phasic cholinergic signals did not make the mice insensitive to threats or incapable of learning threat locations. To identify specific deficits underlying this behavioral change, we investigated how the mice adjusted their path choice after experiencing threats in the probabilistic paradigm. After receiving an air puff on the high-threat path, both groups chose the other path in the subsequent lap starting from the first session (two-way mixed ANOVA, Session, $F(4,56) = 1.054$, p = 0.388; Group, $F(1,14) = 0.131$, p = 0.723; **Figure 6A**, left). On the other hand, after receiving an air puff on the low-threat path, both groups initially chose the other path in the subsequent lap. Over 5 days of training, however, the no-opsin group learned to stay on the low-threat path, whereas the opsin group continued to switch to the high-threat path (two-way mixed ANOVA, Session × Group interaction, $F(4,44) = 3.442$, p = 0.017; follow-up one-way repeated measures ANOVA, No opsin, $F(4,16) = 3.291$, p = 0.038; Opsin, $F(4,28) = 0.579$, p = 0.680; **Figure 6A**, right).

The above analysis revealed that the no-opsin group learned to ignore rare threats on the low-threat path, but the opsin group did not. This observation led us to theorize that the no-opsin

group estimated threat probability by integrating threat occurrences across multiple laps, and this ability might be impaired in the opsin group. To test this idea, we correlated a sequence of path choices against the threat probability estimated with the threat history over different numbers of laps (*Figure 5—figure supplement 2G*). Consistent with our view, the path choice of the no-opsin group showed stronger correlations with the estimated threat probability as the greater number of laps was used to calculate threat probability (*Figure 6B*, left). In contrast, the path choice of most mice in the opsin group was not correlated with the estimated threat probability regardless of the number of laps used to calculate the probability (right). The proportion of mice with significant correlations was greater in the no-opsin group than in the opsin group in both paths (chi-square test, ht-p, $X^2_1 = 4.073$, p = 0.044; lt-p, $X^2_1 = 7.748$, p = 0.005; *Figure 6C*). These observations suggest that abnormally strong cholinergic signals during threats made mice adjust their choices after each threat without tracking its occurrence over time. This would make them highly sensitive to a surprising threat, even when it falls within the expected range of variability.

## Discussion

The mPFC has been implicated in the evaluation of probabilistic outcome contingencies (*St. Onge and Floresco, 2010*; *Paine et al., 2015*; *Zeeb et al., 2015*; *Orsini et al., 2018*) and contains cells sensitive to outcome expectations (*Schoenbaum et al., 1998*; *Wallis and Miller, 2003*; *Padoa-Schioppa and Assad, 2006*; *Kim et al., 2008*; *Sul et al., 2010*) and prediction errors (*Amiez et al., 2006*; *Bryden et al., 2011*; *Hayden et al., 2011*; *Matsumoto et al., 2007*; *Seo and Lee, 2007*; *Hyman et al., 2017*; *Kehrer et al., 2024*). Here, we have shown that this function and coding property are modulated by phasic cholinergic signals time-locked to outcomes.

During our probabilistic spatial learning task, PL cells robustly responded to threats and their omission (*Figure 3A*). From the beginning, PL cells differentiated outcome-related activity for the two threat sites (*Figure 3B*, *Figure 3—figure supplement 1A*), indicating the conjunctive code of outcome and its location. With subsequent learning, omission-evoked activity was strengthened, while threat-evoked activity was unchanged. Notably, such strengthening occurred only for rare threat omissions on the high-threat path but not for frequent omissions on the low-threat path (*Figure 3C*), paralleling the gradual differentiation of threat expectations for these paths (*Figure 1D, F*). Furthermore, we also observed that outcome-evoked responses were sensitive to mice's outcome expectations on each lap (*Figure 4A*). Although patterns of activity differentiation were diverse (*Figure 4B, C*, *Figure 4—figure supplement 1A*), ~10% of PL cells responded to better- and worse-than-expected outcomes. Half of them reversed the response direction (i.e., increased vs. decreased activity) for positive and negative errors, indicating the selectivity for signed prediction errors. Meanwhile, the remaining showed the same response direction for both error types, indicating the selectivity for unsigned prediction errors. These findings align with the selectivity of prefrontal cells for reward prediction errors (*Amiez et al., 2006*; *Bryden et al., 2011*; *Hayden et al., 2011*; *Matsumoto et al., 2007*; *Seo and Lee, 2007*; *Hyman et al., 2017*; *Kehrer et al., 2024*) and expand the critical role of mPFC in prediction error coding to aversive outcomes.

Among these coding properties of PL cells, cholinergic terminal stimulation (1) abolished the learning-dependent increase in the responses to rare threat omission (*Figure 3F*, *Figure 3—figure supplements 1B*) and (2) increased the proportion of cells encoding signed prediction errors (*Figure 4D*). Notably, these effects emerged over days but not across minutes within the first day, suggesting that enhanced cholinergic signals did not immediately change how PL cells responded to subsequent threats. Instead, it modulated the gradual restructuring of outcome representations with learning. An open question is how phasic cholinergic signals achieve such modulation. Cholinergic modulation of the mPFC is highly complex because both nicotinic and muscarinic ACh receptors are expressed in pyramidal cells and interneurons in a layer-dependent manner (*Picciotto et al., 2012*; *Venkatesan and Lambe, 2020*). Indeed, in vitro slice preparations, the application of ACh inhibits pyramidal cells in superficial layers but excites pyramidal cells in deep layers (*Kassam et al., 2008*; *Gulledge et al., 2009*; *Poorthuis et al., 2013*). These mixed effects at least partially explain why our optogenetic stimulation did not result in a drastic increase in PL cells responding to threats (*Figure 2*) or the corner of the maze (*Figure 2—figure supplement 1*).

In parallel to the modulation of cell excitability, ACh also controls the plasticity of glutamatergic synapses (*Verhoog et al., 2016*; *Sabec et al., 2018*) by modulating glutamate release onto pyramidal

cells (*Lambe et al., 2003*; *Wang et al., 2006*) or inhibitory synaptic transmission (*Couey et al., 2007*). One potential function of such modulation is to adjust the relative impacts of various long-range inputs on PL cells. For example, some basolateral amygdala (BLA) cells encode unsigned prediction errors at the time of outcomes (*Belova et al., 2007*; *Roesch et al., 2010*) and outcome-predictive cues (*Klavir et al., 2013*). Notably, activating the M1 subtype of muscarinic ACh receptor induces long-term depression of field excitatory post-synaptic potentials in BLA–PL pathways (*Maksymetz et al., 2019*). Therefore, enhanced cholinergic signals may reduce the influence of BLA inputs carrying unsigned prediction errors via the activation of M1 receptors, thereby increasing the impact of others carrying signed prediction errors, such as dopamine inputs from the ventral tegmental area. Although some controversy exists on how dopamine neurons respond to aversiveness (*Mirenowicz and Schultz, 1996*; *Fiorillo, 2013*), several studies have reported that dopamine neurons are inhibited by unexpected threats (*Matsumoto and Hikosaka, 2009*; *Matsumoto et al., 2016*) and excited by unexpected threat omission (*Salinas-Hernández et al., 2018*). In addition, dopaminergic signals at the time of outcome modulate aversive learning (*Luo et al., 2018*; *Vander Weele et al., 2018*). Investigating such hetero-synaptic plasticity would be a promising future direction to deepen our understanding of intricate cholinergic modulation of prefrontal neural computations.

Alternatively, phasic cholinergic signals may convey signed prediction errors by themselves and directly improve the selectivity of PL cells. However, this is unlikely for several reasons. First, BF cholinergic cells do not differentiate their responses between expected and surprising threats (*Hangya et al., 2015*; *Hegedüs et al., 2023*). Second, cholinergic terminals in the PL were activated by threats but not by surprising threat omission (*Tu et al., 2022*; *Figure 4—figure supplement 2*). These observations suggest that phasic cholinergic signals covey the presence of aversive stimuli to PL cells but not prediction errors.

To decipher specific deficits that emerged from the modified prediction error coding in the PL, we conducted additional behavioral experiments (*Figure 5*) and analyzed mice's choices on a lap-by-lap basis (*Figure 6*). We observed that cholinergic terminal stimulation did not affect the likelihood of correcting the choice after threats on the high-threat path (*Figure 6A*), suggesting that enhanced phasic cholinergic signals do not affect the learning from aversive outcomes. This observation explains why the manipulation did not affect learning in the deterministic paradigm, in which mice needed to correct the choice after every threat (*Figure 5G*). The intact learning also eliminates the possibility that cholinergic terminal stimulation affected the sensitivity to threats, the motivation to obtain the reward, the discrimination between the two paths, and the learning of threat locations. In contrast, the enhanced cholinergic signals reduced the likelihood of staying on the low-threat path after a surprising threat (*Figure 6A*). Also, it decoupled the choice from the estimated threat probability based on threat occurrence across multiple laps (*Figure 6B, C*). Theories link the handling of surprising outcomes to the processing of prediction error signals (*Soltani and Izquierdo, 2019*; *Soltani and Koechlin, 2022*; *Preuschoff and Bossaerts, 2007*). Specifically, when threats occur reliably, the estimate of threat probability should be updated with each prediction error signal to reduce the magnitude of error

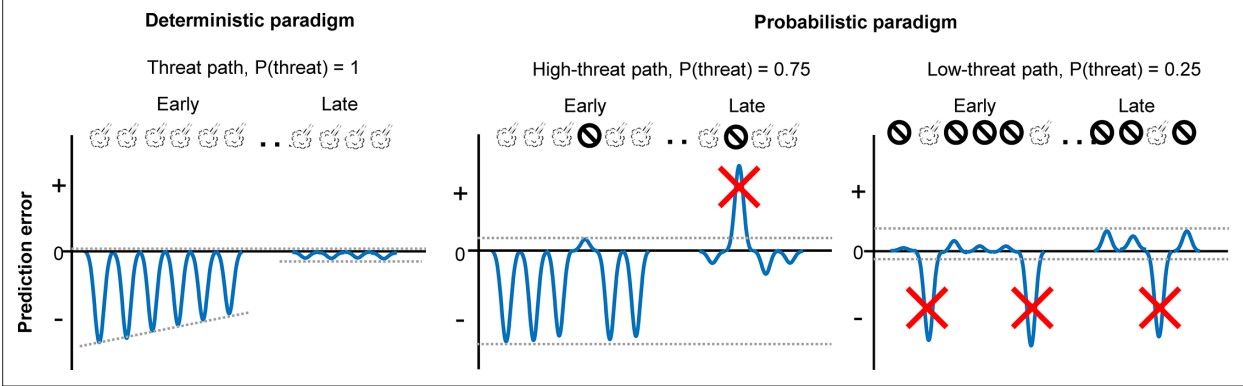

**Figure 7.** Schematic representations of aversive prediction errors in the deterministic and probabilistic paradigms. Aversive prediction errors become positive for unexpected threat delivery while negative for unexpected threat omission (*McNally et al., 2011*; *Iordanova et al., 2021*). Red crosses indicate prediction error signals with abnormally large amplitude. We argue that the role of outcome-evoked phasic cholinergic signals is to suppress learning from these occasional surprising outcomes.

signals (*Figure 7*). In contrast, when threats occur probabilistically, the magnitude of prediction error signals massively fluctuates depending on whether a threat occurred or not on each occasion. With such large variability, the estimate of threat probability should not be updated by error signals with unusually large magnitudes. Also, to assess whether an error signal is an anomaly, one must track the absolute magnitude of error signals over time (*Soltani and Izquierdo, 2019*; *Soltani and Koechlin, 2022*; *Preuschoff and Bossaerts, 2007*). The present behavioral results are consistent with these theories and identify that threat-evoked phasic cholinergic signals serve as an essential input enabling PL cells to properly handle prediction errors for adaptive threat learning.

A question remains open as to whether this function is achieved solely by phasic cholinergic signals within the PL. Although we stimulated cholinergic terminals in the PL, the stimulation could retrogradely activate their cell bodies, leading to additional release of ACh at their terminals in other regions. Indeed, ~75% of PL-projecting cholinergic cells also send axons to the anterior cingulate and orbitofrontal cortex (*Chandler and Waterhouse, 2012*). We, therefore, cannot eliminate the possibility that the coordinated release of ACh in multiple prefrontal regions might also contribute to the detected manipulation effects on PL cell activity and adaptive threat learning.

In conclusion, the evidence from cell activity and behavior suggests that the threat-evoked phasic cholinergic signal modulates the gain of learning from surprising threats by controlling the content of outcome coding in the PL. Our findings extend the cholinergic modulation of cognition to an intricate evaluation process of surprising outcomes and its control over adaptive learning and decision-making.

The average *z*-normalized activity across cells with different types of activity differentiation (mean ± SEM). A cell was selected when its differential activity for threats or their omission was greater than 2 (Up) or smaller than –2 (Down). Some cells differentiated their responses to threats, but not their omission (Threat only). In parallel, other cells differentiated their responses to threat omission but not threats (Omission only).

In our previous study (*Tu et al., 2022*), we conducted photometric recording from cholinergic terminals in the PL while mice associated an auditory conditioned stimulus (CS, 100 ms) with an aversive unconditioned stimulus (US, 100 ms) over a 500-ms stimulus interval. In each session, mice received 125 CS presentations. 80% of the CS was paired with the US (CS–US paired trials), while the remaining 20% of the CS was presented alone (CS-alone trials). These trials were inter-mixed and presented in a random order. Over the course of 10 daily sessions (S1–S10), mice developed conditioned responses to the CS, indicating the formation of CS–US association (*Tu et al., 2022*). From S1, cholinergic terminals were strongly activated by the CS (gray arrows) and the US (red arrows). Due to the slow kinetics of calcium indicator (GCaMP6s), the responses to the US overlapped with those to the CS, resulting in a longer peak latency and longer response duration (orange arrow) in CS–US paired trials than CS-alone trials. With learning, the magnitude of CS-evoked components became larger in both CS–US paired and CS-alone trials. However, the response latency or duration was never extended in CS-alone trials, indicating a lack of components evoked by surprising US omission. Thus, cholinergic terminals in the PL convey the presence of threat and threat-predictive cues, but not surprising threat omission.

## Materials and methods

### Animals

Adult ChAT[ires-Cre] mice (RRID:IMSR_JAX:028861) were bred as homozygotes. Cre recombinase is expressed under the ChAT promotor in these mice without disrupting endogenous ChAT expression. In total, sixty-eight 8- to 20-week-old male ChAT[ires-Cre] mice were used in optogenetics and calcium imaging experiments. Mice were housed under a 12-hr light/dark cycle with ad libitum access to food and water. Seven days before the behavioral testing, food was restricted to maintain 85–90% of free-feeding body weight for the experiments.

### Viral vectors

AAV9-EF1a-DIO-hChR2(H134R)-EYFP-WPRE-HGHpA (RRID:Addgene_20298), AAV9-EF1a-DIO-EYFP (RRID:Addgene_27056), AAV5-Syn-FLEX-rc[ChrimsonR-tdTomato] (RRID:Addgene_62723), and AAV5-FLEX-tdTomato (RRID:Addgene_28306) were purchased from the Addgene.

AAV1-CamKIIa-GCaM6f-WPRE-bGHpA ready-to-image virus (Catalog ID: 1000-002613) was purchased from the Inscopix (Bruker).

## Surgery
### Viral infusion
Eight- to 12-week-old ChAT[ires-Cre] mice were anesthetized (1–2% isoflurane by volume in oxygen at a flow rate of 0.8 l/min; Fresenius Kabi) and administered with meloxicam (20 mg/kg) subcutaneously. Mice were placed in a stereotaxic frame, and holes were made in the skull bilaterally above the BF (+0.2 mm AP, ±1.3 mm ML) (*Paxinos and Franklin, 2019*). A 33 G microinfusion cannula was connected to a 10-µl Hamilton syringe with Tygon tubing. One of the viral vectors (AAV9-EF1a-DIO-hChR2(H134R)-EYFP-WPRE-HGHpA, AAV9-EF1a-DIO-EYFP, AAV5-Syn-FLEX-rc[ChrimsonR-tdTomato], and AAV5-FLEX-tdTomato; 0.8–1.0 µl/side) was infused to the BF (–5.4 mm DV) at a rate of 0.1 µl/min for optogenetic experiments. For imaging experiments, AAV5-Syn-FLEX-rc[ChrimsonR-tdTomato] or AAV5-FLEX-tdTomato (0.8–1 µl/side) was infused to the BF (−5.4 mm DV) at a rate of 0.1 µl/min. Another hole was drilled above the right mPFC (+1.9 mm AP, + 0.35 mm ML). A 200-nl AAV1-CamKIIa-GCaM6f-WPRE-bGHpA was infused to the PL region (−2.2 mm DV) at a rate of 50 nl/min. After infusion, the cannula was left in the brain for 15 min, allowing the viral vectors to diffuse.

### Optic fiber implantation
Three to 6 weeks after viral infusion, mice were placed in a stereotaxic frame as described above. Optic fibers (200-µm core diameter, 0.39 NA; Thorlabs) were implanted bilaterally at 30° lateral to the medial angle above the PL (+1.9 mm AP, ±1.4 mm ML, –1.8 mm DV). The optic fibers were secured to the skull with resin cement (3M dental, Relyx unicem). The mice were given 14 days to recover from the surgery before behavioral testing.

### Gradient index (GRIN) lens implantation
Three to 6 weeks after viral infusion, mice were placed in a stereotaxic frame as described above. A 1.2-mm diameter craniotomy was made above the mPFC (centered at +1.9 mm AP, +0.6 mm ML). A 25 G blunt needle was slowly lowered above the PL (+1.9 mm AP, +0.6 mm ML, −1.5 mm DV) at a rate of 10 µm/s to make a track for GRIN lens implant. Five minutes later, the 25 G blunt needle was slowly removed at a rate of 10 µm/s. A ProView integrated lens (1 mm diameter, 4 mm length; Inscopix) with baseplate attached was implanted 100–200 nm above the PL (+1.9 mm AP, +0.55 mm ML, −1.7 to 1.9 mm DV; lowered at a rate of 2 µm/s). To prevent dust, the lens and basal plate were secured to the skull with resin cement (3M dental, Relyx unicem) and covered by a magnetic cover (Inscopix). The mice were given at least 21 days to recover from the surgery before behavioral testing.

## Behavioral experiments
### Probabilistic spatial learning
The apparatus was a 75 cm × 75 cm square-shaped maze with an 8-cm wide track and 15-cm walls. The inner wall was inclined 30° outward to prevent cables from tangling at the corners. A camera was positioned centrally 1 m above the maze to monitor the mice's movement. At the two opposite corners of the maze, a circle feeding trough (2 cm in diameter) was placed at the rear wall (reward site). A sugar pellet (20 mg, Bio-Serve) was manually delivered to the trough using a Tygon tube accessible through the opening (1 cm diameter) at the rear wall right above the trough. Additionally, two pieces of Tygon tubing (6 mm internal diameter and total length of 20 cm) were attached outside the two sides walls of the other two corners (each side 10 cm long, threat site). The tubes and side walls were perforated with 20 holes × 5 mm diameter. Using plastic Y connectors, these tubes were connected to solenoids and a compressed air cylinder (Whisper-tech). An air puff (300 ms duration, 35 psi) was delivered via this Tygon tubing. During the task, the mouse's head, body, and tail locations were extracted from a video recorded at 30 frames per second by an overhead webcam (Ethovision software, Noldus Information Technology). The entry to the reward or threat sites were defined as the time when the mouse body center entered the pre-defined region of interest (reward site, 20 cm × 20 cm; threat sites, 8 cm × 20 cm), and the corresponding timestamps were stored. In a subset of laps,

an air puff was delivered 100 ms after the entry to a threat site, at which Ethovision software emitted a TTL pulse to activate a solenoid valve connected to the tubes installed at that threat site.

During the first 5 days (pre-training stage), all mice ran on the maze to collect sugar pellets at the reward sites without any air-puff delivery. Subsequently, each mouse underwent one of three paradigms. In the probabilistic paradigm, air puffs were delivered 75% of the time at one threat site and 25% of the time at the other threat site. In the deterministic paradigm, air puffs were delivered 100% of the time at one threat site and 0% of the time at the other threat site. In the random paradigm, air puffs were delivered 50% of the time at both threat sites. Sugar pellets were always delivered in all stages regardless of path choice and air-puff deliveries.

## One-photon single-cell calcium imaging with optogenetic manipulations

Calcium imaging videos were recorded using a head-mounted miniature microscope (miniscope, Inscopix nVoke acquisition system v2.0) and accompanying software (IDAS, v1.7.1). The fluorescence power during imaging sessions ranged from 0.5 to 0.8 mW/mm$^2$ at a rate of 20 Hz, with an excitation wavelength of 455 ± 8 nm. Ethovision software was used to trigger and control the Inscopix Data acquisition (DAQ) box in synchrony with behavioral video tracking (30 fps) using a webcam installed near the ceiling.

Before imaging experiments, mice were handled by the experimenter for 7 days. The mice were further habituated to the miniscope mounting procedure by attaching a dummy scope for two to three sessions (~10 min long). During these sessions, focus depth was adjusted for each animal and was maintained consistently throughout the subsequent imaging sessions. During the same period, these mice were subjected to food restriction to maintain 85–90% of their free-feeding body weight.

The mice underwent the pre-training (5 days), probabilistic (5 days), and random (5 days) stages. To mitigate photobleaching, we only recorded images on a subset of sessions (*Figure 1—figure supplement 1*). A dust cover was removed, and the implanted lens was cleaned with 75% ethanol and double-distilled water before attaching the miniscope to the mouse in each session. The mouse was placed in a cylinder (20 cm diameter, 25 cm height) in the maze's center for 5 min to acclimate to the environment. Subsequently, the cylinder was removed, and the mouse was transferred to one of the reward sites, initiating the session automatically. Each imaging session lasted 40 min and comprised two LED-ON and two LED-OFF epochs. The order of epochs was counterbalanced across mice and sessions (ON-OFF–ON-OFF or OFF-ON–OFF-ON). During the LED-ON epoch, Arduino sent TTL signals to turn on solenoids to deliver air puffs when the mouse entered a threat site. Concurrently, another TTL signal activated an orange LED integrated into the miniscope for 300 ms (20 Hz, 8 mW/mm$^2$, 590–650 nm wavelength). In the LED-OFF epoch, the orange LED was not activated concurrently with air puffs. In a subset of sessions (P2, P4, R2, and R4), the LED stimulation was applied during every air-puff delivery without capturing calcium imaging videos.

## Optogenetic experiments

A separate cohort of mice first underwent the pre-training stage and moved on to the probabilistic or deterministic stage, during which they received bilateral optogenetic manipulations of cholinergic terminals in the PL. Daily sessions lasted 15–20 min or paused when mice ran over 60 laps. Each lap started when a mouse exited one reward site and ended when it entered the other reward site. A blue laser (300 ms, 20 Hz, 3–5 mW, 473 nm wavelength; Laserglow Technologies) was turned on during every air-puff delivery for mice expressing ChR2. An orange LED (300 ms, 20 Hz, 5 mW, 625 nm wavelength; M625F2, Thorlabs) was turned on during every air-puff delivery for mice expressing ChrimsonR. These light stimulation parameters were chosen based on previous works ditional release of A (*Stamatakis et al., 2018*; *Vander Weele et al., 2018*). Although we did not validate the release of ACh in the PL following the optogenetic stimulation, previous works have reported that ACh is released following optogenetic excitation of cholinergic terminals with light stimulation parameters comparable to ours (*Kimchi et al., 2024*; *Gritton et al., 2016*).

## Tissue collection, immunostaining, and image acquisition

Three to 5 days after the last session, mice were subcutaneously administered with avertin (20 mg/kg) and transcardially perfused with 0.9% saline followed by chilled 4% paraformaldehyde (PFA). The brains were immersed in 4% PFA for 48–72 hr, transferred to 30% sucrose/PBS, and stored at 4°C. A series of

50-μm coronal sections were collected from the area containing the mPFC (AP: +1.3 to +2.4 mm) and BF (AP: +0.6 to −0.6 mm) using a cryostat (CM3050S, Leica Biosystems). The tissues were washed in PBS three times and incubated with 10% donkey (# D9663, Sigma-Aldrich, RRID:AB_2810235) or goat serum (# ab156046, Abcam) for 2 hr at room temperature (RT) followed by a three-time wash in PBS.

To check the transgene expression, the BF sections were incubated with primary antibodies against choline acetyltransferase (ChAT; Goat, # AB144P, 1:200, Millipore, RRID:AB_90661) for 24–36 hr at 4°C. Following three times wash in PBS, the brain sections were then incubated with a secondary antibody (Rhodamine-conjugated donkey-anti-goat, # 705-025-147, 1:200, Jackson ImmunoResearch, RRID:AB_2340389) for 2 hr at RT. Finally, the BF sections were mounted using an antifade mountant with DAPI (# P36935, Thermo Fisher Scientific). Images for viral spread and optic fiber track identification were collected using a confocal microscope (Zeiss) under a 10× objective.

## Behavioral analysis

X- and y-coordinates of a mouse's position were extracted from the video tracking data. The timestamps were collected when a mouse entered/exited the threat and reward sites. A lap was counted as a 'full' lap when a mouse exited a reward site, passed a threat site, and entered the other reward site. A lap was counted as a 'half' lap when a mouse exited a reward site, entered a threat site, and returned to the same reward site. If a mouse exited a reward site and returned to the same reward site without entering a threat site, it was not counted as a half lap. Adaptive path choice was defined as the percentage of the laps in which a mouse chose the path associated with a lower threat probability among all full and half laps.

The speed approaching the threat site was calculated as the average speed in a 500-ms window immediately before the threat site entry during each lap. In each session, the median of the approaching speed was used to categorized laps into fast and slow laps (*Figure 1G*). The reaction to outcomes was quantified as the average speed in a 500-ms window immediately after the threat site entry.

Latency to threats was defined as the time difference between the reward site exit and the threat site entry. Latency to reward was defined as the time difference between the threat site exit and the reward site entry. The latency to initiate a subsequent lap was defined as the time in a reward site before initiating a new lap.

To examine if threat presence in a previous lap affected path choice in the subsequent lap, we used the data in the probabilistic stage, in which air puffs were delivered 75% of the time on one path (high-threat path) and 25% of the time on the other path (low-threat path). We first selected laps with air-puff delivery on each path in each session. If the number of these laps was less than two in a session, that session was discarded. We then selected mice with usable data in all five sessions. For each path, we calculated the proportion of laps in which a mouse chose the low-threat path in the subsequent lap.

To investigate how many laps mice used to estimate threat probabilities associated with the paths, we examined the relationship between a lap-by-lap path choice and the threat probability estimated based on the history of threats over different numbers of laps. In each mouse, laps were concatenated across all five sessions of the probabilistic stage. We then coded path choice in each lap as 1 for the high-threat path and 0 for the low-threat path (*Figure 5—figure supplement 2*). We also calculated threat probability based on threat occurrence in the past 1–50 laps. The Pearson correlation coefficient was calculated between the path choice and the estimated threat probability. We then counted the mice that showed a significant correlation with threat probability calculated with any lap number.

## PL imaging data analysis

### Cell extraction

All imaging data were imported into the Inscopix Data Processing Software (IDPS) to spatially down-sample by a factor of 2 and temporally down-sample by a factor of 2. Pre-processed movies underwent spatial bandpass filer (low cut-off = 0.005 pixel-1, high cut-off = 0.500 pixel-1) and motion correction (mean frame) using IDPS. We used an established cell-detection algorithm, constrained nonnegative matrix factorization (CNMFe) in motion-corrected movies for cell detection. Each extracted region of interest was manually checked to select the potential neuronal traces from the noise.

## Outcome-evoked responses

In each cell, $Ca^{2+}$ activity in 3-s windows starting from 1 s before the threat site entry was collected and binned into 100-ms bins. The binned activity was averaged across laps and converted into z-scores with the mean and SD of activity during a 1-s window before the threat site entry (first 10 bins). This normalization procedure was applied to each lap type separately. Specifically, to test the effect of LED stimulation on threat-evoked activity (*Figure 2*), laps were categorized into three types: air-puff omission, air-puff delivery, and air-puff delivery with LED stimulation. To examine how PL cells changed threat- and omission-evoked responses with learning (*Figure 3*), laps were categorized into four types: by which path a mouse took and whether it received air puff or not. To isolate learning-dependent changes in puff-evoked responses from the direct effect of LED stimulation, laps with air-puff delivery and LED stimulation were not included in this analysis. Lastly, to investigate how PL cells differentiated outcome-evoked responses depending on outcome expectation (*Figure 4*), laps were categorized into four types based on how fast a mouse approached the threat site and whether it received air puff or not.

The response magnitude was calculated by averaging z-normalized activity over a 500-ms window starting from the threat site entry. Cells were deemed to respond to threats or their omission (*Figure 3B, E*) if the response magnitude was greater than 2. Cells were deemed to differentiate outcome-evoked responses depending on outcome expectation (*Figure 4C*) if the difference in the response magnitude between surprising and expected outcomes was greater than 2 or smaller than −2.

## Statistical analysis

The sample size was chosen based on previous studies using similar experimental approaches (*Cho et al., 2023*; *Kin et al., 2023*; *Sun et al., 2020*). Animals were randomly allocated to different groups, and all experimenters were blinded to the group assignments. The data were presented as the group mean ± standard error of the mean unless otherwise specified. Statistical analyses were performed with GraphPad Prism, JASP, or MATLAB. To determine the statistical significance, we used one-way (repeated) ANOVA, two-way mixed/repeated measures ANOVA, unpaired t-test, chi-square test, Kolmogorov–Smirnov test, and binominal test. Degrees of freedom were corrected with Greenhouse–Geisser correction when sphericity was violated. For post hoc analyses, Tukey's multiple comparison test was used following one-way repeated measures ANOVA. Significance was defined as *p < 0.05, **p < 0.01, ***p < 0.001.

## Acknowledgements

This work was supported by NSERC Discovery Grant, CFI Leaders Opportunity Fund (Kaori Takehara-Nishiuchi), and NSERC graduate fellowship (Gaqi Tu).

## Additional information

### Funding

| Funder | Grant reference number | Author |
| --- | --- | --- |
| Natural Sciences and Engineering Research Council of Canada | | Kaori Takehara-Nishiuchi |
| Canada Foundation for Innovation | | Kaori Takehara-Nishiuchi |
| Natural Sciences and Engineering Research Council of Canada | CGSD3-547178 | Gaqi Tu |

The funders had no role in study design, data collection and interpretation, or the decision to submit the work for publication.

## Author contributions

Gaqi Tu, Conceptualization, Data curation, Formal analysis, Funding acquisition, Investigation, Visualization, Methodology, Project administration, Writing - review and editing; Peiying Wen, Adel Halawa, Data curation, Methodology; Kaori Takehara-Nishiuchi, Conceptualization, Data curation, Formal analysis, Supervision, Funding acquisition, Investigation, Visualization, Writing - original draft, Writing - review and editing

## Author ORCIDs

Gaqi Tu ⓘ https://orcid.org/0000-0001-5807-0798
Kaori Takehara-Nishiuchi ⓘ https://orcid.org/0000-0002-7282-7838

## Ethics

All methodological procedures were conducted under the regulation of the University of Toronto Animal Care Committee and the Canadian Council on Animal Care (AUP20012042).

Reviewer #1 (Public review): https://doi.org/10.7554/eLife.102986.2.sa1
Reviewer #2 (Public review): https://doi.org/10.7554/eLife.102986.2.sa2
Reviewer #3 (Public review): https://doi.org/10.7554/eLife.102986.2.sa3
Author response https://doi.org/10.7554/eLife.102986.2.sa4

# Additional files

## Supplementary files

Supplementary file 1. Summary of the number of imaged cells in each mouse.

MDAR checklist

## Data availability

Data generated or analyzed during this study are included in the manuscript and supporting files; source data files have been provided for Figures 1D, F, 5D, G, 6A, and Figure 5-figure supplement 2A-F.

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
