## [Editor Report · eLife Assessment]

This is an **important** study using a combination of optogenetics and calcium imaging to provide insight into the function of the cholinergic input to the prelimbic cortex in probabilistic spatial learning as it relates to threat. These data are timely in contributing to an ongoing discussion in the field about the role of phasic cholinergic signaling to the cortex, about which relatively little is known. The strength of the evidence is **incomplete** and could be improved by changes in task design and analyses, cross-validation of the conditions in calcium imaging, as well as the incorporation of control experiments to more definitively show it is indeed acetylcholine working in this circuit.

---

## [Referee Report · Reviewer #1 (Public review)]

Tu, Wen, et al. investigated the activity of mPFC putative glutamatergic neurons during a probabilistic threat discrimination and avoidance learning task using miniaturized GRIN lens implantation and single-photon calcium imaging in freely moving mice. In conjunction with this cellular recording, they employed channelrhodopsin-mediated optogenetic excitation of terminals from basal forebrain cholinergic projection neurons coupled to the delivery of an air puff on either of two maze paths with differential threat probability. The authors found that the optogenetic manipulation altered mPFC encoding of outcomes and disrupted animals' behavioral adaptation. Over the course of multiple learning sessions, optogenetically stimulated mice lagged behind control animals in resolving the differential threat probabilities on the two paths and making adaptive choices. In particular, the animals with optogenetic stimulation of cholinergic terminals were significantly more likely to switch to the path with higher threat probability after having just gotten a rare air puff on the generally "safer" path. Combined with data from a deterministic version of the task showing that optogenetically stimulated mice could behaviorally discriminate between the paths appropriately under such circumstances, these results suggest an impairment in the experimental animals' ability to make use of threat history over multiple trials. This comparison of probabilistic and deterministic versions of the same task is a highlight of this paper, representing a thoughtfulness about what information can be gleaned from such variations in the design of behavioral experiments that is all too often lacking. These data are timely in contributing to an ongoing discussion in the field about the role of phasic cholinergic signaling to the cortex, about which relatively little is known.

While the ensemble recording of mPFC neurons during the task appears to be reliable and well-designed and the behavioral effects of the optogenetic stimulation are convincing, some major weaknesses of the paper limit its usefulness to others in the field:

(1) Optogenetic excitation of presynaptic terminals can lead to antidromic action potentials that alter the firing properties of the target cell (see the excellent review on challenges of and strategies for presynaptic optogenetic experiments Rost et al., Nat Neurosci 2022). To their credit, the authors explicitly acknowledge this fact, but they believe that the only alternative possibility is that their intervention could lead to increased acetylcholine release at collateral projections in other prefrontal subregions. In fact, we do not know that the mechanism mediating the behavioral changes observed involves acetylcholine at all, as many ChAT+ basal forebrain neurons co-transmit using GABA (Saunders et al., Nature, 2015; Saunders et al., eLife, 2015; Granger et al., Neuropharmacology, 2016). A very useful internal control, which is recommended by Rost et al. for such presynaptic excitation experiments, would be to locally infuse nicotinic or muscarinic cholinergic antagonists into the mPFC in an attempt to reverse the optogenetically induced deficit; this would resolve whether the effect is indeed mediated by cholinergic neurotransmission and if it is specific to the mPFC.

(2) In a similar vein, the fact that LED illumination in the no-opsin control group appears to increase activity in prefrontal neurons (Figure 2C) and, moreover, has a functional effect in disrupting location-selective cellular activity to a similar extent as in the ChrimsonR group (Figure S3) is inadequately explained and cause for concern. Although the authors argue that the degree or "robustness" of puff-evoked activity was significantly greater in the ChrimsonR group as compared to fluorophore-only controls, their statistical test for demonstrating this is the Kolmogorov-Smirnov test (Figure 2D), thus showing that the two samples likely are drawn from different distributions but little else.

(3) Throughout the paper, the authors rely heavily on the Kolmogorov-Smirnov and binomial tests (Figures 2D, 3, 4D, S3, S4) to compare distributions in this manner, but it is unclear to me why these would be the most appropriate statistical tests for what they seek to demonstrate. Given the holistic nature of these tests in comparing the shape and spread of distributions, I am concerned that they might be inflating the significance of the differences between groups. Even if the authors were seeking a nonparametric statistical test, which most likely would be quite appropriate, there are nonparametric versions of ANOVA that they could use (e.g. Kruskal-Wallis, Friedman). Indeed, in much of this data set a repeated measures statistical analysis would seem to be called for, whereas the Kolmogorov-Smirnov test assumes that the two samples must be independent of each other. The most notable example of this premise being violated is in Figure 3, where data from the same cell populations in the same animals are being compared between experimental days and across various trial types.

---

## [Referee Report · Reviewer #2 (Public review)]

Summary:

The authors tested:

(1) Whether mice learn that they are more/less likely to receive an aversive air puff outcome at different corners of a square-shaped open field apparatus, under 75%/25% probabilistic contingencies;

(2) Whether stimulating basal forebrain cholinergic neurons and terminals in the prefrontal cortex affects learning in this context; and

(3) Whether stimulating cholinergic neurons affects prefrontal cortical single neuron calcium signaling about outcome expectations during learning and contingency changes. They found that mice that received cholinergic stimulation approached high and low aversive outcome probability sites at similar velocities, while control mice approached high probability sites slower, suggesting that cholinergic stimulation impaired learning. Cholinergic stimulation reduced cortical neuron calcium activity during trials on the high-probability corner when the outcome was not delivered. The authors provide additional characterization of cellular responses during delivery/omission trials in high/low probability corners, using running speed as a proxy for low versus high expectations. The study will likely be of interest to those who are interested in prediction and error signaling in the cortex; however, the task and analyses do not permit very easy or clear dissociation of prediction versus prediction error signaling and place field versus place field-expectation multiplexing. The study has several strengths but some weaknesses, which are discussed below.

Strengths:

It is clear the authors were very careful and did a great job with their image processing and segmentation procedures. The details in the methods are appreciated, as are the supplemental descriptive statistics on cell counts.

There are careful experimental controls - for example, the authors showed that the effects of cholinergic stimulation with air puff present are greater than without it, thus ruling out effects of stimulation on cellular physiology that were independent of learning or the task.

The addition of a channelrhodopsin stimulation group is helpful to show that the effects are robust and not wavelength/opsin-specific.

The prefrontal cortex cholinergic terminal stimulation experiment is a great addition. It shows that the behavioral effects of cell body stimulation, which was used in the imaging experiments, are similar to cortical terminal stimulation, where the imaging was performed.

Weaknesses:

The analyses were a bit difficult to follow and therefore it is difficult to determine whether the cells are signaling predictions versus prediction errors - a very important distinction.

The task does not fully dissociate place field coding, since learning about the different probabilities necessarily took place at different areas in the apparatus. Some additional analyses could help address this.

---

## [Referee Report · Reviewer #3 (Public review)]

Summary:

Using a combination of optogenetic tools and single-photon calcium imaging, the authors collected a set of high-quality data and conducted thorough analyses to demonstrate the importance of cholinergic input to the prelimbic cortex in probabilistic spatial learning, particularly pertaining to threat.

Strengths:

Given the importance of the findings, this paper will appeal to a broad audience in the systems, behavioural, and cognitive neuroscience community.

Weaknesses:

I have only a few concerns that I consider need to be addressed.

(1) Can the authors describe the basic effect of cholinergic stimulation on PL neurons' activity, during pretraining, probabilistic, and random stages? From the plot, it seems that some neurons had an increase and others had a decrease in activity. What are the percentages for significant changes in activities, given the intensity of stimulation? Were these changes correlated with the neurons' selectivity for the location? If they happen to have the data, a dose-response plot would be very helpful too.

(2) Figure 2B: The current sorting does not show the effects of puff and LED well. Perhaps it's best to sort based on the 'puff with no stim' condition in the middle, by the total activity in 2s following the puff, and then by the timing in the rise/drop of activity (from early to late). This way perhaps the optogenetic stimulation would appear more striking. Figure 3Aa and Ba have the same issue: by the current sorting, the effects are not very visible at all. Perhaps they want to consider not showing the cells that did not show the effect of puff and/or LED.

Also, I would recommend that the authors use ABCD to refer to figure panels, instead of Aa, Ab, etc. This is very hard to follow.

(3) The authors mentioned the laminar distribution of ACh receptors in discussion. Can they show the presence/absence of topographic distribution of neurons responding to puff and/or LED?

(4) Figure 2C seems to show only neurons with increased activity to an air puff. It's also important to know how neurons with an inhibitory response to air-puff behaved, especially given that in tdTomato animals, the proportion of these neurons was the same as excitatory responders.

(5) Page 5, lines 107 and 110: Following 2-way ANOVA, the authors used a 'follow-up 1-way rmANOVA' and 'follow-up t-test' instead of post hoc tests (e.g. Tukey's). This doesn't seem right. Please use post hoc tests instead to avoid the problem of multiple comparisons.

(6) Figure 1H: in the running speed analysis, were all trials included, both LED+ and LED-? This doesn't affect the previous panels in Figure 1 but it could affect 1H. Did stimulation affect how the running speed recovers?

On a related note, does a surprising puff/omission affect the running speed on the subsequent trial?

(7) On Page 7, line 143, it says "In the absence of LED stimulation, the magnitude of their puff-evoked activity was reduced in ChrimsonR-expressing mice...", but then on line 147 it says "This group difference was not detected without the LED stimulation". I don't follow what is meant by the latter statement, it seems to be conflicting with line 143. The red curves in the left vs right panels do not seem different. The effect of air puff seems to differ, but is this due to a higher gray curve ('no puff' condition) in the ChrimsonR group?

(8) Did the neural activity correlate with running speed? Since the main finding was the absence of difference in running speed modulation by probability in ChrimsonR mice, one would expect to see PL cells showing parallel differences.

---

## [Author Response]

(1) We do not know that the mechanism mediating the behavioral changes observed involves acetylcholine at all. (Reviewer 1)

The reviewer rightly pointed out the co-release of acetylcholine (ACh) and GABA from cholinergic terminals. We believe that the detected behavioral changes are because of the augmentation of this innate mixed chemical signal. We agree that identifying the receptor specificity is an essential next step; however, addressing this point requires a currently unavailable research tool to block cholinergic receptors for a few hundred milliseconds. This temporal specificity is vital because acetylcholine is released in the medial prefrontal cortex (mPFC) on two distinct timescales, the slow release over tens of minutes from the task onset and the fast release time-locked to salient stimuli (TelesGrilo Ruivo et al., 2017). Moreover, the former slow signal is far more robust than the latter phasic signal. The pharmacological experiments suggested by the reviewer will suppress both the tonic and phasic signals, making it difficult to interpret the results. Given the rapid technological advancement in this field, we hope to investigate the underlying mechanisms in detail in the future.

(2) It is unclear whether mPFC cells are signaling predictions versus prediction errors. (Reviewer 2)

As the reviewer pointed out, mPFC cells signal the prediction of imminent outcomes (Baeg et al., 2001; Mulder et al., 2003; Takehara-Nishiuchi and McNaughton, 2008; Kyriazi et al., 2020).

However, the key difference between prediction signals and prediction error signals is their time course. The prediction signals begin to arise before the actual outcome occurs, whereas the prediction error signals are emitted after subjects experience the presence or absence of the expected outcome. In all our analyses, cell activity was normalized by the activity during the 1-second window before the threat site entry (i.e., the reveal of actual outcome; Lines 655-659). Also, all the statistical comparisons were made on the normalized activity during the 500-msec window, starting from the threat site entry (Lines 669670). Because this approach isolated the change in cell activity after the actual outcome, we interpret the data in Figure 4C as prediction error signals.

(3) The task does not fully dissociate place field coding. (Reviewer 2)

The present analysis included several strategies to dissociate outcome selectivity from location selectivity (Figure 4). First, we collapsed cell activity on two threat sites to suppress the difference in cell activity between the sites. Second, our analysis compared how cell activity at the same location differed depending on whether outcomes were expected or surprising (Figure 4C). Nevertheless, we can use the present data to investigate the spatial tuning of mPFC cells. Indeed, an earlier version of this manuscript included some characterizations of spatial tuning. However, these data were deemed irrelevant and distracting when this manuscript was reviewed for publication in a different journal. As such, these data were removed from the current version. We are in the process of publishing another paper focusing on the spatial tuning of mPFC cells and their learning-dependent changes.

(4) The basic effects of cholinergic terminal stimulation on mPFC cell activity are unclear. (Reviewers 1, 3)

We acknowledge the lack of characterization of the optogenetic manipulation of cholinergic terminals on mPFC cell activity outside the task context. As outlined in the discussion section (Lines 309-321), cholinergic modulation of mPFC cell activity is highly complex and most likely varies depending on behavioral states. In addition, because we intended to augment naturally occurring threatevoked cholinergic terminal responses (Tu et al., 2022), our optogenetic stimulation parameters were 3-5 times weaker than those used to evoke behavioral changes solely by the optogenetic stimulation of cholinergic terminals (Gritton et al., 2016). Based on these points, we validated the optogenetic stimulation based on its effects on air-puff-evoked cell activity during the task (Figure 2C, 2D).

(5) Some choices of statistical analyses are questionable (Reviewers 1, 3)

We used the Kolmogorov-Smirnov (KS) test to investigate whether the distribution of cell responses differed between the two groups (Figure 2D) or changed with learning (Figure 3Ac, 3Bc). As seen in Figure 3Aa, some mPFC cells increased calcium activity in response to air-puffs, while others decreased. We expected that the manipulation or learning would alter these responses. If they are strengthened, the increased responses will become more positive, while the decreased responses will become more negative. If they are weakened, both responses will become closer to 0. Under such conditions, the shape of the distribution of cell response will change but not the median. The KS test can detect this, but not other tests sensitive to the difference in medians, such as Wilcoxon rank-sum tests. In Figure 2D, KS tests were applied to the independently sampled data from the control and ChrimsonRexpressing mice. In Figure 3Ac and 3Bc, we used all cells imaged in the first and fifth sessions. Considering that ~50% of them were longitudinally registered on both days, we acknowledge the violation in the assumption of independent sampling. In Figure 1D, we detected significant interaction between the group and sessions. Several approaches are appropriate to demonstrate the source of this interaction. We chose to conduct one-way ANOVA separately in each group to demonstrate the significant change in % adaptive choice across the sessions in the control group but not the ChrimsonR group. The cutoff for significance was adjusted with the Bonferroni correction in follow-up paired t-tests used in Figure 1F.